# Microfluidic live tracking and transcriptomics of cancer-immune cell doublets link intercellular proximity and gene regulation

Bianca C. T. Flores [1,2,3,9,15], Smriti Chawla[4,15], Ning Ma[1,10,15], Chad Sanada[5,11], Praveen Kumar Kujur[1,12], Rudy Yeung[5,13], Margot B. Bellon [1], Kyle Hukari[5], Brian Fowler[5], Mark Lynch[5,14], Ludmilla T. D. Chinen[2], Naveen Ramalingam[5✉], Debarka Sengupta [4,6,7,8✉] & Stefanie S. Jeffrey [1✉]

Cell–cell communication and physical interactions play a vital role in cancer initiation, homeostasis, progression, and immune response. Here, we report a system that combines live capture of different cell types, co-incubation, time-lapse imaging, and gene expression profiling of doublets using a microfluidic integrated fluidic circuit that enables measurement of physical distances between cells and the associated transcriptional profiles due to cell–cell interactions. We track the temporal variations in natural killer—triple-negative breast cancer cell distances and compare them with terminal cellular transcriptome profiles. The results show the time-bound activities of regulatory modules and allude to the existence of transcriptional memory. Our experimental and bioinformatic approaches serve as a proof of concept for interrogating live-cell interactions at doublet resolution. Together, our findings highlight the use of our approach across different cancers and cell types.

[1] Department of Surgery, Stanford University School of Medicine, Stanford, CA 94305, USA. [2] Circulating Tumor Cells Group, A.C.Camargo Cancer Center, São Paulo, SP 01508-010, Brazil. [3] Cancer Biology and Epigenetics Group, IPO Porto Research Center (CI-IPOP), Portuguese Oncology Institute of Porto (IPO Porto), 4200-072 Porto, Portugal. [4] Department of Computational Biology, Indraprastha Institute of Information Technology, New Delhi 110020, India. [5] New Technologies Group, Fluidigm Corporation, South San Francisco, CA 94080, USA. [6] Department of Computer Science and Engineering, Indraprastha Institute of Information Technology, New Delhi 110020, India. [7] Centre for Artificial Intelligence, Indraprastha Institute of Information Technology, New Delhi 110020, India. [8] Institute of Health and Biomedical Innovation, Queensland University of Technology, Brisbane, QLD 4001, Australia. [9] Present address: Department of Clinical Genetics, Lillebaelt Hospital, Vejle, Denmark. [10] Present address: Akoya Biosciences, Menlo Park, CA 94025, USA. [11] Present address: Systems Integration Group, Inscripta Inc, Pleasanton, CA 94588, USA. [12] Present address: Cancer Biology Laboratory, School of Life Sciences, Jawaharlal Nehru University, New Delhi 110067, India. [13] Present address: Seer Inc., Redwood City, CA 94065, USA. [14] Present address: BioSkryb Genomics, Inc., Durham, NC 27713, USA. [15] These authors contributed equally: Bianca C. T. Flores, Smriti Chawla, Ning Ma. ✉email: naveen.ramalingam@fluidigm.com; debarka@iiitd.ac.in; ssj@stanford.edu

Cell–cell communication sustains the multicellular organism as an integral unit via direct physical interactions, surface receptor-ligand interaction, cell signaling from adjacent cells, nearby cells, or even distant organs[1,2]. The investigation of cell–cell interaction in the tumor microenvironment (TME) is one of the barriers to understanding cancer progression and identifying new therapeutic targets[3]. Despite the advances in high-throughput microscopy and single-cell based molecular analysis, tools to precisely quantify live cell–cell interactions are lacking or, more recently, characterized using combinations of spatial -omics techniques on fresh-frozen or formalin-fixed complex tissues and live cell analyses[4–8]. We developed a microfluidic workflow involving capture and co-incubation of live single stromal/cancer cells or doublets using the single-cell dosing mRNA-seq integrated fluidic circuit (IFC) system (Fluidigm®), which provides both spatial and transcriptional cell–cell interactions. To demonstrate the performance for the quantification of the cell–cell interaction, we applied our platform for natural killer (NK) - triple-negative breast cancer (TNBC) cancer-immune doublets (CIDs).

TNBC was chosen as a model due to its aggressive nature and lack of estrogen receptor, progesterone receptor, and human epidermal growth factor receptor 2 that limit the use of targeted therapies[9,10]. However, promising responses have been seen with immunotherapy[11], which is emerging as an important component of cancer treatment[12,13]. Clinically promising discoveries exploit specific features of tumor-immune cell crosstalk involving immunosuppression and anti-tumoral response signaling[14–17]. Among immune cells, NK cells can efficiently kill multiple neighboring cells with oncogenic transformation of surface markers[18,19]. NK cell activation, combined with their capacity to enhance antibody responses, supports NK cells' role as anticancer agents[20]. It is speculated that genetically engineered endogenous NK cells can exert tumor immunosurveillance and influence tumor growth[21,22]. However, heterogeneity is ubiquitous in human cancer making the selection of personalized treatment/therapy a challenge. It is expected that NK cell heterogeneity further contributes to NK-tumor crosstalk dynamics with differential modulation of their cytotoxic response, triggering tumor death when the balance between activation and inhibitory protein levels are considered[23–25]. It is essential, therefore, to characterize the molecular level cues emanating from single NK cells when they encounter tumor cells. Thus, a single-cell platform for NK-cancer cell interaction measurement is much needed to study NK-cancer immunotherapy.

To better understand NK-tumor cell interactions, we present a microfluidic workflow involving capture and co-incubation of single NK and cancer cells (CIDs) using the Polaris™ Single-Cell Dosing mRNA Seq IFC (Fluidigm)[26] (Fig. S1). Traditional sequencing methods can only identify cell populations, analyzing the average of the signals within each group of cells, but the heterogeneity in tumor cells cannot be deciphered. Single-cell sequencing technologies can reliably detect the heterogeneity among cells. Further, in doublets of cells, it is possible to find specific markers and correlate them with the anti-tumor response demonstrated by each doublet[27].

The doublets captured in the microfluidic chambers were tracked for cell–cell distances, using time-lapse imaging. After 13 h of incubation with the exchange of growth medium at a defined interval of time (5 h), the cells were subjected to single-cell RNA sequencing (scRNA-seq). This offered a total of 290 transcriptomes, including single NK and cancer cells as control and NK-cancer cell doublets (CIDs). Unsupervised clustering analysis of the scRNA-seq expression profiles revealed heterogeneity in the TNBC cell line. We also observed a small number of NK killing events among the co-incubated CIDs, which

allowed us to characterize the gene expression signature associated with NK-mediated lysis event of the companion cancer cells. We correlated the hourly computed cell–cell distances with terminally profiled gene expression vectors. We noted the existence of transcriptional memory, governed by precise regulatory modules active in a time-bound manner. Interestingly, we found the genes driving cancer-immune cell distances to have longer half-lives. In addition, we investigated the protein pairs involved in cellular communication, which provided cues for the inflated activity of CD24/SIGLEC10 and ANXA1/EGFR in the cancer-NK doublets and supporting a previously described interaction between CD24/SIGLEC10 as a potent immunotherapy target for ovarian and triple negative breast cancer[28].

## Results

To study the interactions between NK and TNBC cells, we captured single NK-92MI cells (interleukin-2 independent Natural Killer cell line), single MDA-MB-231 cells, and cancer-immune doublets (CIDs, one NK-92MI cell and one MDA-MB-231) and incubated the cells for 13 h using the Fluidigm Polaris system[26,29–32] (Figs. 1a, S1). Time-lapse images of CIDs were captured every hour to measure the distance between the CIDs. Following incubation, single cells and doublets were processed for RNA-sequencing. The integrated fluidic circuit can perform on-chip multi step chemistry that includes cell capture, co-incubation, lysis, reverse transcription, and cDNA amplification[30]. Single-cell and CIDs were analyzed for expression of marker genes that distinguished the two cell types. We identified previously known markers for both NK-92MI cells and MDA-MB-231 cells. In the case of NK cells, we observed strong expression of NK cell marker genes KLRD1, LAIR1, CCR6, and TNFRSF9. Single cancer cells showed TNBC marker genes HMGA1, ANKRD11, and TACSTD2 (Fig. 1b) when analyzed using SCANPY[33] toolkit for analyzing single-cell gene expression data. Further to validate our cell type annotations, we took advantage of the imaging capability of the Polaris and Leica systems. Prior to cell selection on the microfluidic IFC, the NK and cancer cells were stained with CellTracker™ Deep Red Dye and CellTracker™ Orange CMRA Dye respectively. The z-score normalized intensities of NK and cancer cell channels were subjected to 2D visualization (plotted using R package scatterD3) revealing the grouping of cells as per their annotations based on cell staining (Fig. S2). Unsupervised clustering with Seurat v3 and associated Uniform Manifold Approximation and Projection (UMAP) based 2D visualization[34] of the transcriptomes revealed two separate clusters, which were primarily dominated by clonal heterogeneity of the cancer cell line (Fig. 1c). The single NK cells shared one of the clusters (Fig. S4), cluster 1, with a cancer cell sub-group. Upon performing the principal component analysis (PCA) exclusively on the transcriptomes from this cluster, we noted spatial segregation of the NK cells (Fig. 1c). To further delineate the cancer cell heterogeneity, we performed unsupervised clustering of single cancer cells separately, which also resulted in two distinct clusters (Fig. 1d), featuring differentially expressed genes (Fig. S3).

**Distance tracking of cancer-immune cell interactions over time shows transcriptional memory.** We tracked CIDs for dynamic changes in the distance between cancer and the companion NK cells under incubation in the same microfluidic chamber over 13 h (Fig. 1a). At the end of the 13th hour, transcriptome sequencing was performed on the CIDs (n = 102). This allowed us to determine the association between terminally measured gene expression with cancer/immune cell distances measured across different time points. Transcriptomes profiled at the end of the 13th hour featured transcripts that correlate significantly with

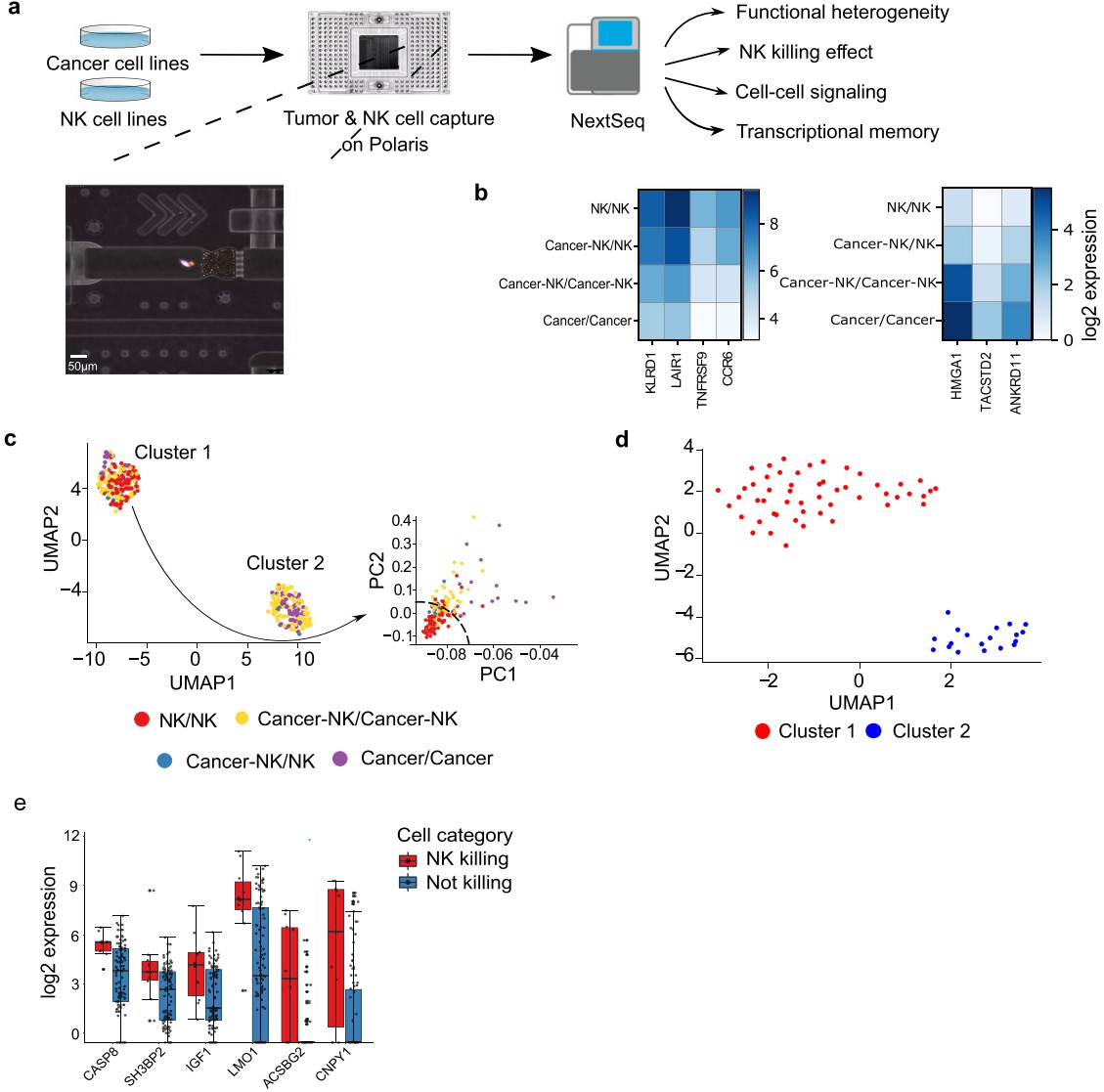

**Fig. 1 Schematic workflow for cell interaction studies. a** The workflow involves cancer and NK cells propagated in culture and stained off-chip, which are then captured as live single cells and doublets, incubated/co-incubated and imaged over time. Inset shows a microscopic image of a chamber on the microfluidic Integrated Fluidic Circuit (IFC) containing an elongated MDA-MB-231 (pink) and round NK cell (red) doublet. The cells are then lysed, reverse transcribed, and the cDNA amplified in situ within the chambers. Down-stream library preparation, sample barcoding, and sequencing are performed off-chip using the Illumina NextSeq system. **b** Heatmap plotted using Scanpy showing average expression of marker genes for single NK and cancer cells, confirming their lineage identity. **c** UMAP-based visualization of the cells shows two separate clusters mainly due to cancer cell line heterogeneity (cancer cells in purple). A further dimension reduction of cluster 1 using PCA shows the separation of cancer (purple) and NK cells (red). Legend key indicates cell status at the beginning and the end of the time-course tracking (e.g., cancer-NK/NK denotes interactions which initially consisted of both cancer and the NK cells in a chamber, and subsequently the NK cell remained in the chamber after co-incubation due to a cell killing event). **d** UMAP based visualization of single cancer cells showing heterogeneous populations of cancer cell lines. **e** Boxplots showing selected differentially expressed genes distinguishing killing vs non-killing events. Parts of (**a**) were created using an icon from FreeImages.com/olagosta.

CID distance measurements across all the time points (Fig. 2a, b). Surprisingly, we identified time-bound activities of at least three distinct gene modules (Fig. 2b). Mechanistically, we postulated that the transcripts would preferentially stem from slowly degrading genes. To test this, we considered mRNA half-life estimates in K562 cells for more than 5000 genes[35]. We compared the distribution of half-life estimates of transcripts with ones that were not part of the gene modules. A higher average degradation rate was observed for the transcripts (Welch Two Sample t-test, $P$-value = 0.04867), substantiating our conjecture (Fig. 2c). At time point 6 (after 5 h incubation and a change in culture medium), we noted a new set of genes (module 2; M2) expressed. However, we did not observe a similar shift in gene expression at

later culture medium change time points. We performed module wise transcription factor activity analysis using ShinyGO[36] and RcisTarget[37], which inferred the regulatory role of three potential transcription factors, namely BRCA1, YY1, and THAP1. Among these, BRCA1 is predicted by ShinyGO, whereas YY1 and THAP1 along with their candidate target genes by RcisTarget (Fig. 2d).

The microfluidic system allowed us to design experiments that tracked NK cell killing events and the associated transcriptomic signatures. We observed ten NK cell killing events across a total of 132 CID interactions. Differential expression analysis between CIDs subgroups showed unique gene expression signatures between NK killing and non-killing events. Among the 187 upregulated genes in the NK killing group were *CASP8*, *SH3BP2*,

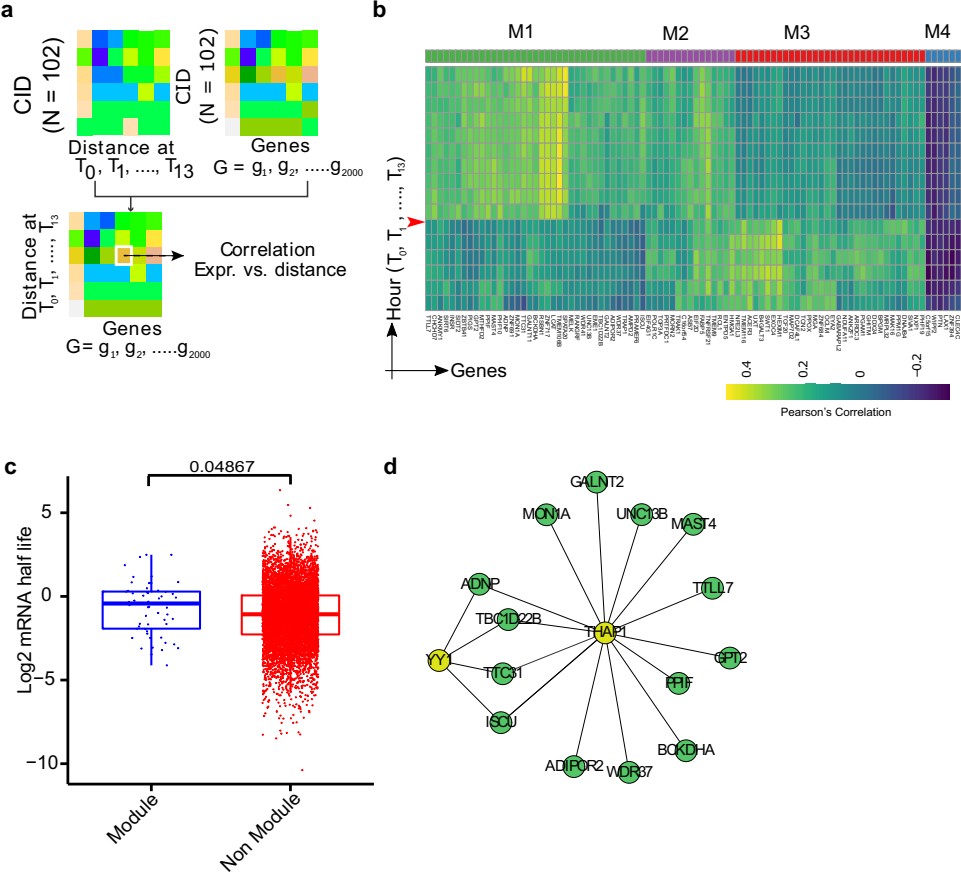

**Fig. 2 Schematic workflow to estimate the correlation between cancer cell-NK cell (CIDs) distance and terminal expression profile of CIDs. a** The correlation analysis takes two matrices as input - the first one, containing cell–cell distance across different time points, and the second one contains the gene expression measurements across the CIDs. **b** Heatmap shows correlation between cell–cell distances and terminal CID gene expression profiles. Genes showing strong association with the cell–cell distances are clustered into four groups based on the correlation patterns (M1 to M4). The heatmap reveals that physical distances between NK and cancer cells affect gene expression profiles, which get carried forward to define future cellular states. The time point of first cell culture medium exchange is shown by a red arrow. This was labeled T5/T5' and T10/T10' to denote images taken before and immediately after media exchange for a total of 16 images that portray distances T0 to T13. The color scale relates to high (yellow) and purple (low) Pearson's correlation. **c** Boxplot showing the distribution of mRNA half-life (hours) of genes belonging to the module as shown in the heatmap (Fig. 2b) and genes not belonging to these modules. We observed a higher median value of mRNA half-life for the genes belonging to the modules (Welch Two Sample t-test, *P*-value = 0.04867). **d** Transcription Factor binding motif Enrichment Analysis of M1 specific genes and annotations of these motifs to TFs using RcisTarget identifies THAP1 and YY1 as potential transcriptional regulators.

*IGF-1*, *CNPY1*, and *LMO1* (Fig. 1e). We applied PROGgene V2, a tool for prognostic implications measurement of genes[38], to the Cancer Genome Atlas (Breast Invasive Carcinoma; TCGA-BRCA patients), and analyzed the impact of these 187 upregulated genes on overall survival. 164 out of 187 upregulated genes overlapped with TCGA-BRCA dataset from PROGgene V2 (Supplementary Data 1). We observed a subtle survival advantage in the patients having a higher mean expression of this combined gene signature (Hazard ratio (HR) = 0.11, *p* < 0.05) (Fig. 3).

## Discussion

TNBC is a highly aggressive form of breast cancer with limited targeted treatment options. However, immunotherapy has recently turned out to be a promising strategy in the clinical management of the disease. In addition to current widely used T cell-mediated immunotherapy, research is also focused on NK cells that perform key roles in innate immune response in cancer. As such, comprehensive characterization of interacting NK and cancer cells at single-cell resolution might unravel actionable biomarkers and pathways involved in tumor growth and progression.

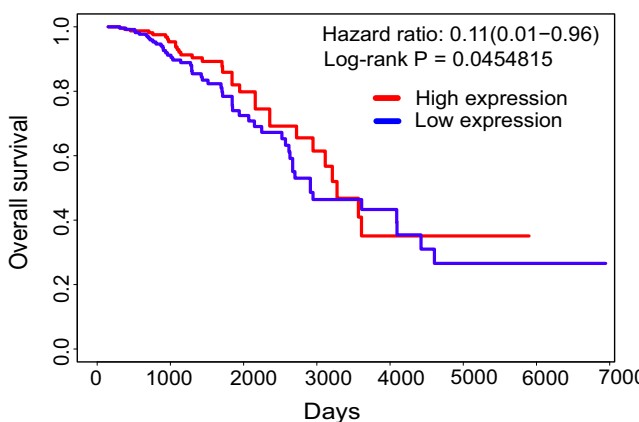

**Fig. 3 Kaplan–Meier overall survival analysis of TCGA breast cancer (BRCA) dataset.** A gene signature associated with NK-killing events was used to stratify the TCGA BRCA dataset (*n* = 594) in terms of overall survival. PROGgene V2 was used for this analysis.

Here, we performed gene expression analysis of NK-cancer cell doublets resulting from the interacting distance between doublets measured over time. Statistical and bioinformatic investigation of the data enabled the identification of gene expression signatures associated with successful and unsuccessful cancer cell lysis (killing events). Using TCGA gene expression profile data, we further confirmed that the identified signatures are linked to patient survival. Using single cell analysis, we also noted that the heterogeneity of the two cancer subclones confounded the process of dimension reduction by overshadowing the distinction of the NK cells' identity.

We noted activation of exclusive, exposure time-bound, gene regulatory modules governing cell–cell physical proximity during the life cycle of NK-cancer cell interaction under co-incubation. We also noted an association between CID cell distance and their terminally profiled transcriptome in live cells. Similarly, Gide and colleagues reported a potential association between cancer/immune cell proximity with anti-PD-1 therapy response in melanoma patients[39]. This underscores the importance of cell–cell distance as an informative parameter to understand immunosurveillance and response in cancer. We observed changes in gene expression modules over time that correlated to the distance between live cells, suggesting the existence of transcriptional memory. Recent work has highlighted the controlled synthesis and degradation of mRNA transcripts as a major regulatory strategy influencing cell fate decisions[40]. Transcriptional memory is a phenomenon that allows cells to retain reversible memory to respond to similar stimuli encountered in the future[41]. At time point 6 (after a 5 h incubation and a change in culture medium), we noted a new set of genes expressed. During the initial hours of co-incubation, the cells might have undergone the formation of lytic synapses as reported previously[1,42]. which possibly might be responsible for the change in gene expression, as noted at T5 timepoint in Fig. 2b.

In our study, we observed a regulatory role of three potential transcription factors, namely *BRCA1*, *YY1*, and *THAP1*, in transcriptional memory. *BRCA1* is a well-studied tumor suppressor gene and has known implications in breast and ovarian cancers. Among the remaining two, *YY1* promotes oncogenic activities in breast cancer[43], whereas *THAP1* plays a key role in DNA repair and is also found to be overexpressed in breast cancers[44]. One of the genes from module 1 is *TRAP1*, whose overexpression is involved in promoting breast tumor growth. On the other hand, it also suppresses metastasis by regulating mitochondrial dynamics[45]. Another gene *MELK* from this module promotes TNBC proliferation. Targeting *MELK* can result in cell cycle arrest by reducing cyclin B1 and increasing p27 and p-JNK[46]. *EYA2* is also involved in promoting breast cancer proliferation. Its overexpression results in an increase of proliferative markers cyclin E, PCNA, and EGFR[47].

Cell–cell signaling is a major component of cancer-immune cell interactions. We used gene expression as a surrogate for protein-protein interactions, focusing on interactions involved in cellular communication as featured in the iTALK database[48].

To estimate the extent of these protein activities, we computed Pearson's correlation coefficient for iTALK-featured protein pairs across CIDs (taking advantage of having the gene expression of both cell types together) and the other single cells (Fig. 4). We observed an elevated correlation between *ANXA1* and *EGFR* in CIDs. ANXA1 is involved in mediating endocytosis of the EGFR receptor ANXA1/S100A11 complex[49–51], and mediates cell–cell communication via exosomal EGFR[52]. We also observed a higher correlation between *CD24/SIGLEC10* in the CIDs. Similarly, higher coordination between *EGFR* and *HSP90AA1* was observed in CIDs. A similar pattern was observed in the case of *CD24/SIGLEC10* transcripts, suggesting a potential coordination

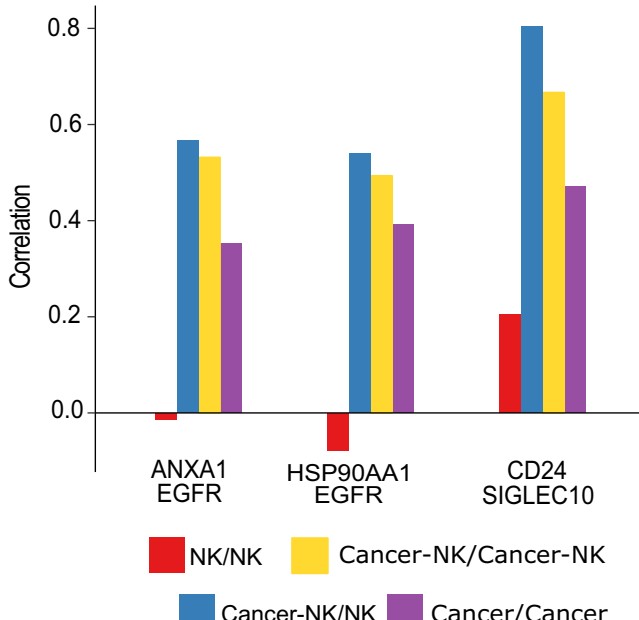

**Fig. 4 Inference of protein-protein associations by gene expression profiling.** Barplots showing increased correlation values among CIDs for three gene pairs *ANXA1/EGFR, HSP90AA1/EGFR,* and *CD24/SIGLEC10.* NK = Natural Killer Cells. Legend key indicates cell status at the beginning and the end of the time-course tracking.

between the two. A past study reported the association of SIGLEC10 in impeding NK cell function and poor patient survival in hepatocellular carcinoma (HCC); CD24/SIGLEC10 interaction may thus be involved in regulating NK cell function[53]. In another study, the possibility of CD24/SIGLEC10 interaction in TNBC under the exposure of tumor-associated macrophages (TAMs) was reported. It has been observed that CD24 and SIGLEC10 are overexpressed in different tumor types and the TAMs, respectively[28]. Targeting this interaction may be therapeutically important. Another transcript pair observed was *EGFR/HSP90AA1*. HSP90AA1 is critical for maintaining the stability and function of its client protein EGFR. This stabilization promotes pathogenesis in breast, head, and neck cancer[54,55]. This occurs via EMT and tumor migration activating signaling pathways in MDA-MB-231 cells[56].

When the CIDs were analyzed for cell killing events, we noted a gene expression signature that included *CASP8, SH3BP2, IGF-1*, and *LMO1*. *CASP8*, when activated through *FASLG*, results in activation of the extrinsic pathway of apoptosis in the target cells[57]. On the other hand, *SH3BP2*[58] and *IGF-1*[59] have shown to play a decisive role in NK cell development and cytotoxicity. A strong differential expression signal was observed for several genes that are largely undocumented for their NK cytotoxicity links. These include the overexpression of *LMO1* (overexpressed in T lymphocytes from lymphoblastic leukemia)[60], *CNPY1* and *ACSBG2* in the NK-killing group, which has not been described in NK cells till today.

Currently, it is difficult to co-incubate single cells in a controlled environment that simultaneously allows the study of physical interactions between live cells and its effect on gene expression[61]. Here we report a cell–cell interaction study using an automated environment that precisely controls temperature, humidity, gas composition, and media exchange to continuously monitor and measure the distance between cells. By processing the doublets on-IFC (in the same chambers) for cell lysis, reverse transcription, and cDNA amplification using a microfluidic

multi-step chemistry, proximity measurements can be directly linked to downstream transcriptomics changes.

The use of this microfluidic system enabled the identification of unique features of NK cells' anti-cancer activities. Highly coordinated gene expression profiles were identified as a result of dynamic changes in the physical distance of interacting NK and cancer cells. Upon further investigation, these transcripts were confirmed to have overall higher RNA half-lives, reinforcing transcriptional memory as a key regulatory strategy of cells. We could also trace increased coordination among some specific transcript pairs, manifested through gene expression readouts. In the future, this microfluidic workflow could provide new leads when studying immuno-oncology cellular interactions, which may be considered while developing and administering NK cell-based immunotherapies.

## Methods

**Culturing MDA-MB-231 and NK-92MI.** MDA-MB-231 cells (triple-negative breast carcinoma, ATCC® HTB-26™) were cultured using the DMEM (high modified) culture medium supplemented with 10% inactivated fetal bovine serum and 1% penicillin/streptomycin, replicating the culture every three days, keeping the culture at a confluence of 40% at the time of passage, at 37 °C in a humid atmosphere at 5% $CO_2$. The adherent cells were dissociated with the TrypLE reagent (Gibco) and resuspended in the complete culture medium.

NK-92MI cells (human NK cells genetically modified to produce interleukin 2, ATCC® CRL-2408) were cultured using the AMEM (Alpha Minimum Essential medium) culture medium, plus 0.2 mM inositol, 0.1 mM mercaptoethanol, 0.02 mM folic acid, 12.5% fetal bovine serum and 12.5% horse serum. The cells were homogenized to separate the clusters of NKs prior to the replication of the culture into a new 25 cm² flask (1 mL of cells cultured previously + 9 mL new culture medium). Both cell lines were cultured separately, tested and confirmed negative for mycoplasma, for the subsequent cell–cell interaction experiment using the Polaris system.

Both cell lines were cultured in separate plates, as described above. The co-culture was performed after the doublets were selected by the Polaris system for 13 h incubation.

**Selection and incubation using the Polaris system (Fluidigm).** The first step is to prime the IFC with beads that will allow the adhesion of the cells to be incubated in the microchambers. After treatment, reagents and cells already labeled using specific markers to differentiate cell types (for NK cells, celltracker far red (CTFR) and; and celltracker orange (CTO) for cancer cells) and viability (calcein AM) were pipetted into the IFC. The selection of the cells to be incubated was performed on the Polaris system. The system was configured to select NK (CTFR + calcein AM) and cancer cells (CTO + calcein AM).

Every IFC had wells containing up to 12 single cancer cell controls, up to 12 single NK cell controls, and up to 24 cancer-NK doublets. After selection, the equipment was programmed to keep the cells in incubation for 13 h, performing the replacement of culture medium (20% DMEM + 80% AMEM) every 5 h (timepoints 5 and 10), and time-lapse images every hour, including images captured before and immediately after culture medium change (timepoints 5/5' and 10/10').

71 single cancer cells, 77 single NK cells, and 132 cells in doublets (NK + cancer cells) were incubated, followed by the cell lysis, reverse transcription, and cDNA amplification. Subsequently, sequencing of the single-cell RNA, using NextSeq (Illumina) was performed.

**Distance measurement between cells.** The distance shown in our study is the shortest distance between the membrane of MDA-MB-231 cell and NK cell on IFC within the doublets (NK + cancer cells) group. The distance has been measured for 13 time points. To analyze the videos frame by frame to yield the distance data, we used ImageJ, a public domain Java-based image processing software developed at the National Institutes of Health[62].

**Data preprocessing.** In total, we obtained expression profiles of 336 cells. Out of the 336 cells, we removed 46. The cells discarded include: 4 empty chambers (no cells could be traced in the chambers from the start to the end); 8 empty chambers that initially contained cancer cells; 10 empty chambers that initially contained NK cells; 2 CIDs that started with single NK cells; and 22 cancer cells that started as CIDs. After removal of these cells, we were left with 290 cells/doublets. The final cell counts are as follows. 1. single NK cells ($n = 77$); 2. single cancer cells ($n = 71$); 3. CIDs throughout all time points ($n = 132$); and 4. CIDs that were left with NK cells alone at the terminal time point ($n = 10$). Further, we discarded cells having less than 2000 expressed genes (non-zero RNA-seq by Expectation Maximization (RSEM) expression value[63]). Next, we retained genes having RSEM expression >5

in at least 10 cells. After these filtering steps, we were left with an expression matrix constituting 290 cells and protein-coding 8907 genes.

**Batch correction, clustering, and visualization of the single cells.** The matrix obtained after performing the basic pre-processing steps was used as input for the Seurat, single cell analysis R package, as well as for other downstream analyses. We used Seurat's data integration workflow to process the data with the genes detected in at least 5 cells, further employing the standard routines for log-normalization, variance stabilizing transformation for identification of highly variable genes using NormalizeData() and FindVariableFeatures() with default parameter settings. The cells used in this study originated from two independent runs. For integration and batch correction, we used the FinIntegrationAnchors() with k.filter=100 for identification of anchor cells that represent matching cell pairs across the two datasets in order to project the transcriptomes into a shared space and IntegrateData() function was used for integrating these anchors, which involve Canonical Correlation Analysis (CCA). These steps provided the batch corrected matrix. 2D map of the cells was created using the RunUMAP() function.

**Differential gene expression analysis.** Limma-voom[64] was used to identify differentially expressed genes (DEGs) among the cell-groups. Top DEGs that qualified adjusted $P$-value cutoff of 0.05 and absolute $\log_2$ fold change cutoff of 1 were further analyzed for their biological significance.

**Survival analysis based on the upregulated genes governing NK cell anti-tumor activity.** PROGgene V2[38] was used for gene signature-based overall survival analysis and the TCGA BRCA dataset, containing survival information for 594 patients. Out of 187 upregulated genes associated with NK killing events, 164 genes overlapped with the TCGA BRCA dataset. PROGgene V2 produced a Kaplan–Meir plot considering, as input, a gene signature comprised of these 164 genes. Divergent survival patterns were observed across high- and low-risk patient groups using the median value as cut-off.

**Regulation of cell–cell distance and transcriptional memory.** We considered two matrices to track the association between temporally recorded cell–cell distance and terminally profiled gene expression. First, the processed gene expression matrix of dimensions $|G| \times |C|$, where $|G|$, and $|C|$ denote the number of genes and the number of CIDs, respectively. Second, the cell–cell distance matrix of dimensions $|C| \times |T|$, where $|T|$ denotes the time points when the cell–cell distance measurements were recorded. We used these two matrices to compute the third matrix of dimensions $|G| \times |T|$ containing the Pearson's correlation coefficients $\rho_{g,t}$ for each gene-time point pairs $\{(g, t)| \ t \in T, g \in G\}$. Notably, $|G| = 2000$, $|C| = 102$, and $|T| = 13$. We retained 90 genes with $|\rho_{g,t}| \geq 0.25$ for at least one time point for further analysis. These 90 genes were subjected to hierarchical clustering based on the computed correlation values. 4 gene modules were retrieved using cutree(). For each of these modules, motif enrichment and transcription factor activity analyses were performed using RcisTarget[37] and ShinyGO[36], respectively. RcisTarget discerned enriched TF binding motifs and reported a module-specific list of TFs, using the hg19-tss-centered-10 kb-7species.mc9nr.feather database containing genome-wide ranking for the motifs. Gene regulatory networks were constructed using igraph[65].

**Tracking mRNA half-life of the genes associated with the modulation of cell–cell distance.** We collected mRNA half-life information for >5000 genes from the study published by Blumberg et al.[35]. Notably, the mRNA half-life measurements were performed on K562 cells. Out of the previously identified 90 cell–cell distance modulating genes, we could find mRNA half-life for 57 genes with an average half-life of 1.05 h.

**Cell–cell signaling among CIDs.** We used iTALK R package featuring information on 2648 cancer-specific transcript interactions. We applied the rawParse() function to select the top 50% highly expressed genes from our processed gene expression matrix (constituting 290 cells and 8907 genes), using mean as a method of choice. This identified 230 gene pairs using the FindLR() function. We computed Pearson's correlation coefficient between expression values associated with the selected pairs across CIDs (cancer-NK/cancer-NK, cancer-NK/NK) and qualified ones that exhibited Pearson's correlation coefficient > 0.4. To rule out the possibility that the co-expression stems solely from either NK or cancer cells alone, we tracked Pearson's correlation coefficient among NK/NK and Cancer/Cancer cases as well. With these criteria, 20 gene pairs were selected, and three of these were found directly involved in breast cancer signaling.

**Statistics & reproducibility.** All the statistical analyses were performed using R software (v4.0.0). The Result and the Method sections provide details on the statistical analyses conducted in this study. We used Limma-voom to estimate the statistical significance of gene expression differences among the cell groups for differential gene expression analysis. For the correlation analyses, Pearson's correlation coefficient was computed. The statistical significance between mRNA half-

lives of genes belonging to the modules vs non-module was calculated using the Welch Two Sample t-test. Survival analysis was performed using the PROGgene V2 tool and compared through a log-rank test.

**Reporting summary**. Further information on research design is available in the Nature Portfolio Reporting Summary linked to this article.

## Data availability

All raw and processed sequencing data generated in this study have been submitted to the NCBI Gene Expression Omnibus (GEO; https://www.ncbi.nlm.nih.gov/geo/) under accession number GSE181591. Source data for figures is available in Supplementary Data 2.

## Code availability

Code can be found at https://github.com/SmritiChawla/NKCell (https://doi.org/10.5281/zenodo.7246721)[66].

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

## Acknowledgements

N.M. and P.K.K. were supported in part by the John and Marva Warnock Research Fund. S.S.J. was supported in part by the Stanford Catalyst for Collaborative Solutions. B.C.T.F. acknowledges the support of the Brazilian National Council for Scientific and Technological Development - CNPq; the Coordination for the Improvement of Higher Education Personnel (CAPES) (grant 88881.190291/2018-01) and the Coordination of A.C.Camargo - Antônio Prudente Foundation. D.S. acknowledges the support of the ihub-Anubhuti-iiitd Foundation set up under the NM-ICPS scheme of the DST. D.S. and S.C. acknowledge the INSPIRE faculty grant [DST/INSPIRE/04/2015/003068] awarded to D.S. from the Department of Science & Technology, India.

## Author contributions

Study conception & design: B.C.T.F., C.D.S., M.L., N.R., and S.S.J. Microfluidic system development and support: C.D.S., R.Y., K.H., B.F, and N.R. Performed experiment or data collection: B.C.T.F., N.M., C.D.S., P.K.K., M.B.B. Computation & statistical analysis: S.C., N.M., N.R., D.S. Data interpretation & biological analyses: B.C.T.F., S.C., N.M., N.R., D.S., and S.S.J. Manuscript writing, review & editing: B.C.T.F., S.C., N.M., N.R., D.S., and S.S.J. with input from all other authors. Supervision: L.T.D.C., N.R., D.S., and S.S.J.

## Competing interests

K.H. and N.R. are employees and stockholders of Fluidigm Corporation. M.L. is a former employee and stockholder of Fluidigm Corporation. D.S. is a stockholder of CareOnco Biotech Pvt. Ltd. and GenterpretR Inc. S.S.J. serves as a scientific advisor for Quantumcyte and Ravel Biotechnology. D.S. is an Editorial Board Member for *Communications Biology*, but was not involved in the editorial review of, nor the decision to publish this article.
