## [Peer Review File · Communications Biology]

Reviewers' comments:

Reviewer #1 (Remarks to the Author):

This manuscript described a method to investigate the relationship of physical distances between cells and the transcriptional profiles under cell-cell interactions. The results suggested that dynamic changes in the physical distance of interacting NK and cancer cells result in highly coordinated gene expression profiles. What's more, proximity measurements can be directly linked to downstream transcriptomics changes. This microfluidic workflow may become an attractive approach to studying complex immune-tumor cellular interactions. The following are comments on this manuscript.

1. Why UMAP-based visualization of four groups of cells in Fig 1 only shows two separate clusters rather than four? And why TU/TU and TU-NK/TU-NK groups exhibit in the same cluster? Did they share any similarities?
2. What are the possible reasons for the difference of gene expression profiles in the physical distance of interacting NK and cancer cells? Is it the difference in the intensity of direct or indirect interaction of NK cells?
3. Cell-cell communications usually involve different numbers of cells in vivo, so is it possible to monitor the relationship of specific numbers of cells to better understand NK-tumor cell interactions?
4. Did the authors ever compare the results of single culture and co-culture, will they exhibit a difference in kinetics and gene expression?
5. Did the authors ever examine tissue models? I wonder whether this approach can be used to capture inherent cell-cell interaction in tissues instead of dissociating single cells?
6. The results of transcriptional memory are interesting, and I wonder how can the phenomenon be reversed and whether it can be used to control the function of cells?
7. What is the throughput of this platform now? And how can it improve to achieve high-throughput analysis?
8. In the text, there are some mistakes and missed information, such as:
(1) In page 2, line 39, "for interrogating live cell interactions at doublet resolution" should be changed to "for interrogating live-cell interactions at doublet resolution".
(2) Please check all scale bars in the manuscript, because some of them were missed, like in Figure 1(c), Figure S1.

Reviewer #2 (Remarks to the Author):

The authors present a proof of concept for a novel approach of investigating the result of interactions between NK-cell/TNBC cell pairs. Both cell pairs as well as single cells are captured in microfluidics chambers and cultured for 13 hours. During this period the cells are continuously monitored via microscopy to determine their distance and potentially cell-contact. Cells are identified using CellTracker staining, and viability is monitored using Calcein AM. After 13 hours, microscopy is used to determine whether killing took place before cells are lysed and single-cell RNA sequencing is performed on either single cells or cell pairs as a whole. The sequencing data allowed them to differentiate the transcriptomes of killing and non-killing cell pairs and correlate these to the monitored distance. This resulted in the determination of; 1) interesting clusters (modules) of genes based on cell-cell distance, 2) transcriptional regulators of interest, 3) genes related to NK-killing events which were also found in the TCGA cohort, 4) cancer specific receptor-ligand pairs. The technique the authors propose is intriguing and very relevant to uncover single-cell heterogeneity within immune interactions. And although it has potential to amount into a useful technique, my opinion on the current state of the manuscript is that it has not yet reached the requirements needed for publication in Nature Communications Biology. Nevertheless, rewriting, which includes more

elaborate explanations of the methods, addition of (already required) data, and more honest/critical discussion of limitations, might raise the manuscript to a level of quality which justifies publication. Below I will underline four major and several minor changes which in my opinion are necessary to raise the manuscripts scientific quality.

Major changes:

- The authors indicate that the presented approach is a proof of concept. This would however require much more information and data to provide insight on how cells behave inside the microfluidic device. The first important factor is how the viability is maintained in the device, Calcein AM is incorporated so this is data that already exists. It would be very interesting to see if all cells maintain viability, including the single cells. Only then certainty can be provided if apoptosis is due to interaction instead of culture conditions, also this would underline why media change after every 5 hours is necessary. In addition to that some information should be provided on how killing events are characterized. The data processing part of the methods indicates that 10 CID`s resulted in NK cells being alone at the final timepoint. However also for 8 and 10 cases a single tumor or NK cells had disappeared, can the media change have influence on this? Additionally, some data that is indispensable is the spatial information of the cells. Are the cells moving around? Can actual cell-cell attachment be identified based on microscopy images? It would greatly benefit the manuscript if a figure can be dedicated to image analysis of cell movement/distance/interaction.
- The correlation made by the authors between distance and gene expression as depicted in Figure 2B does not seem logical. Methods explain that a GxC and TxC matrix is converted into a GxT matrix. However, the gene expression is terminally profiled, and thus has no correlation with the individual distances at all 13 timepoints. The different colors as depicted in the heatmap per gene over time are depicting the correlation between the distance and gene expression. But as gene expression is the terminal value and not measured at each individual timepoint, the color might have well have depicted only the distance, because differences between timepoints are only based on distance. This also raises the question if the change at timepoint 5h (red arrow) is only due to cell movement because of media change. Additional microscopy pictures to determine or rule out that this is the case should be added. It seems much more logical to take the cell distance over time together and identify different "movement patterns" eg; "no contact" or "not contact \diamond contact \diamond release " or "no contact \diamond contact". Or to correlate the gene expression to the duration of cell contact. Also the authors should elaborate some more on the type of interactions they are looking for, be it juxtacrine or paracrine interactions.
- The authors indicate that they find hints towards the existence of transcriptional memory. This can of course be highly interesting for NK-target interactions. However, their elaboration on this is at this point too vague. Transcriptional memory as described in line 179 is "a phenomenon that allows cells to retain reversible memory to respond to similar stimuli encountered in the future" however this suggests that the NK cells have been in contact with the tumor cells before. The paper explains that only 1:1 pairing of cells is performed, and thus no serial-killing-like events can occur. In the 1st section of the methods something is mentioned about NK cell activation, which might suggest co-culture before the cell pairing. But this is not clearly described at all, and the activation data (flow cytometry data) of these NK cells is missing and should be added. The authors should elaborate much more precisely why they think transcriptional memory is occurring, and at which timepoint by which event it is initiated.
- The lysis and sequencing of interacting cell-pairs is incredibly interesting and holds great potential for understanding heterogeneity in immune interactions. However, it also poses enormous technical challenges, and makes it virtually impossible to determine which sequence came from which cell. In Figure S2 the authors nicely illustrate that annotations as found from microscopy data indeed correspond with gene signatures as found in the sequencing. However, it makes drawing conclusion based on discovery of genes much more difficult. In line 184-194 the authors describe three potential transcription factors as found in CID`s that might be of interest. These are all factors that are commonly found to be overexpressed in breast cancer cells, making it logical that they are found in NK-breast cancer cell pairs. The reasoning why these targets are thus interesting should be much more elaborate and consider which cell pairs have had cell-cell contact and which have not. Otherwise, these are just loose observations without conclusions. The same holds for the receptor-ligand interactions as depicted in Figure 4. If the receptor is expressed in cell 1 and the ligand is expressed in

cell 2 then logically sequencing of a pair of these cells will provide a high correlation between the two. These kinds of limitation should be addressed by the author in the discussion. Additionally, the figures describing these results can be more elaborate. eg; A side by side comparison of BRCA1, THAP1 and YY1 expression per module, the results from the ShinyGO analysis next to the RCisTarget analysis, statistical analysis between "TU-NK/TU-NK" and "TU-NK/NK" in Figure 4 to determine if the result is actually due to a killing event, and plotting of the number of reads of both the ligand and receptors that are depicted in Figure 4.

Minor changes:

- Line 83: could the authors elaborate if heterogeneity was also observed in the single NK-cells as it was with the single tumor cells?
- Figure 1C: By eye it seems as if cluster 1 primarily consists of NK/NK events and cluster 2 of primarily TU/TU events. Whilst the two types of CID events are distributed between. Can the authors provide per cluster the exact composition in a separate bar graph?
- Figure 1C and Figure S3A: both describe a cluster 1 and cluster 2, the article suggests that division of both clusters are due to tumor heterogeneity, however, have both UMAP plots been mad with the same data set? Or is the UMAP in Fig S3A made with only the TU/TU events? Cluster 1 in Fig S3A seems to contain much more TU/TU events than cluster 1 in Fig 1C. Can the authors describe this more clearly to rule out the fact that these are just two different clustering dimensions based on different sets of events?
- Can the authors elaborate on how media change at t=5h cause a change in cell-cell distance and how at t=10h it does not? Perhaps by using microscopy images?
- Can the authors elaborate on how media change can potentially affect paracrine signaling?
- Can the authors add representative microscopy pictures for CIDs in different modules? To show that these modules indeed correspond to different cell distance patterns.
- Can the authors elaborate on the "subtle survival advantage" as observed in Figure 3? The P value is above 0.05, perhaps they can provide the actual value?
- Line 180: authors claim that a new set of genes is expressed. This can not be claimed if gene expression is only measured at the terminal timepoint. Can the authors elaborate on this more?
- Line 181-182: Lytic synapses are hypothesized, does the obtained viability and cell-cell distance data not allow to identify these events? Or at least make them more likely?
- Line 183: could the authors elaborate on what they mean by interactions taking a drift?
- Figure S1: T0 is not visible at all, and it is not clear what is to be depicted there. Also, the Leica image is high-resolution, could these be cropped and zoomed to the cell pairs? And include images for several different CIDs?

RESPONSE TO COMMENTS – Reviewer 1

Reviewer #1 (Remarks to the Author):

This manuscript described a method to investigate the relationship of physical distances between cells and the transcriptional profiles under cell-cell interactions. The results suggested that dynamic changes in the physical distance of interacting NK and cancer cells result in highly coordinated gene expression profiles. What's more, proximity measurements can be directly linked to downstream transcriptomics changes. This microfluidic workflow may become an attractive approach to studying complex immune-Cancer cellular interactions. The following are comments on this manuscript.

1. Why UMAP-based visualization of four groups of cells in Fig 1 only shows two separate clusters rather than four? And why CANCER/CANCER and CANCER-NK/CANCER-NK groups exhibit in the same cluster? Did they share any similarities?

Response. We really appreciate your insightful comment. Please note that we have changed Tumor/Tu to Cancer in the manuscript. Citation numbers in this reply to reviewers document refer to references at the bottom of this document.

UMAP-based visualization: We obtained two, rather polarized clusters instead of four (four types of single cells/doublets as discussed in the manuscript) since the intra cluster cell similarities are much more dominant as compared to inter-cluster cell similarities as evident within the individual clusters. The extent of domination was so high that some local segregations are barely visible. As such we demonstrated spatial segregation of NK/NK and Cancer-NK/Cancer-NK in a separate inset PCA plot (**Figure 1**) of cluster 1 alone. The subtle difference between NK/NK (red) cells cluster differently from Cancer-NK/Cancer-NK (green) cells. For cluster 2, please refer to the explanation in the next paragraph. We have also provided bar plots indicating cluster wise cell-type frequencies (**Figure 2**). It can be noted from these plots that CANCER-NK/CANCER-NK look either like NK/NK (share cluster 1) or look similar to CANCER/CANCER (share cluster 2).

CANCER/CANCER and CANCER-NK/CANCER-NK groups exhibit in the same cluster: A plausible explanation for CANCER/CANCER and CANCER-NK/CANCER-NK cells segregating into the same cluster is due to major differences in cell size and total RNA content (pg/cell) of the two cell types (NK-92MI and MDA-MB-231). NK (immune) cells are inherently small in size (~ 8-9 μm size. Reference: (1) with ~ 0.75 pg total RNA/cell. Reference: LIT000037 Rev A Sample Preparation Tips for Single Cell Gene Expression. 10X genomics) while compared to the large size of MDA-MB-231(Cancer) cells (~ 19 μm size. Reference: (2) with ~ 14 pg/cell for similar adherent cell type HEK-293. Reference: LIT000037 Rev A Sample Preparation Tips for Single Cell Gene Expression. 10X genomics). Note that mRNA molecules comprise about 1-5% of total RNA. Hence the contribution of the number of mRNA molecules from NK cells in Cancer-NK is relatively small, until the NK cells are activated, at which time the mRNA molecules increase with

an associated increase in NK cell size. However, even activated NK cells are smaller than MDA-MB-231 cancer cells.

Did they share any similarities: To further understand any similarities between CANCER/CANCER and CANCER-NK/CANCER-NK, we performed differential gene expression analysis between cancer and cancer-NK groups that share cluster 1, and were unable to detect any substantial differences in gene expression programs; genes that were present in the immune cells were also moderately present in cancer-NK doublet (**Figure 3**). In addition, pathway analysis using EnrichR of 36 upregulated genes in cancer from cancer vs doublets indicate the role of PI3K/mTOR signaling (**Figure 4**) that is active in triple negative breast cancer (3).

Figure 1. UMAP-based visualization of the cells shows two separate clusters mainly due to Cancer cell line heterogeneity (Cancer cells in purple). A further dimension reduction of cluster 1 using PCA shows the separation of Cancer (purple) and NK cells (red). Legend key indicates cell status at the beginning and the end of the time-course tracking (e.g., CANCER-NK/NK denotes interactions which initially consisted of both Cancer and the NK cells in a chamber, and subsequently the NK cell remained in the chamber after co-incubation due to a cell killing event).

Figure 2. Barplots indicating cluster wise cell-type frequencies clusters. Legend key indicates cell status at the beginning and the end of the time-course tracking (e.g., CANCER-NK/NK denotes interactions which initially consisted of both Cancer and the NK cells in a chamber, and subsequently the NK cell remained in the chamber after co-incubation due to a cell killing event).

Figure 3. Gene expression comparison of doublets and single cancer cells. The differential gene expression is not substantial in this plot.

Figure 4. EnrichR analysis for 36 genes upregulated in the case of Cancer for Cancer vs Doublets differential gene analysis shows the role of PI3K/mTOR signaling that is active in triple negative breast cancer.

- [1] Dickinson AJ, Meyer M, Pawlak EA, Gomez S, Jaspers I, Allbritton NL. 2015. Analysis of sphingosine kinase activity in single natural killer cells from peripheral blood. *Integr Biol (Camb)* 7: 392-401

- [2] Connolly S, McGourty K, Newport D. 2020. The in vitro inertial positions and viability of cells in suspension under different in vivo flow conditions. *Sci Rep* 10: 1711
- [3] Costa RLB, Han HS, Gradishar WJ. 2018. Targeting the PI3K/AKT/mTOR pathway in triple-negative breast cancer: a review. *Breast Cancer Res Treat* 169: 397-406

2. What are the possible reasons for the difference of gene expression profiles in the physical distance of interacting NK and cancer cells? Is it the difference in the intensity of direct or indirect interaction of NK cells?

Response. We are thankful to the reviewers for pointing this out.

Possible reasons for the difference of gene expression profiles in the physical distance of interacting NK and cancer cells: A relationship exists between cell-cell signaling and distance between them. Difference in type of signaling (paracrine vs juxtacrine) is expected to impact gene expression in the doublets. Note that median half-life of mRNA in a typical carcinoma cell line is ~10 hours (4), so it's expected that some mRNA molecules may persist up to 20 hours. Therefore, traces of genes (molecules) that may regulate cell migration and cell-cell distance at an early time point may still be left at the endpoint.

To support a plausible hypothesis that genes belonging to different modules, shown in **Figure 2 of our manuscript**, have higher overall median half-lives, we analyzed a published dataset from another group that used metabolic labeling for estimating mRNA half-lives across a number of cancer cell lines (5). In this analysis, we obtained mRNA half-life of >5000 genes from earlier published study by Blumberg, Amit, et al. in K562 cells (5) and we observed that the median half-life of genes governing cellular distances was higher than genes not belonging to the modules as depicted in the boxplot below (**Figure 5**). As a result, some of the genes that correlate with distance over the various time windows are expected to have higher mRNA half-life. In other words, mRNA half-life can be thought of as a cell's memory, which hints to its past activities.

Figure 5. Median genes half-life. Median half-life of genes governing cellular distances from the module was higher than genes not belonging to the modules. (Welch Two Sample t-test, p-value = 0.04867)

Is it the difference in the intensity of direct or indirect interaction of NK cells: Possibly both.

Some of the cells are in direct contact with each other at some time points while some are not (i.e. if the distance between two cells is 0, then they are touching; otherwise, they are not touching). We infer that the distance between the cells is directly related to the communication (NK activation and/or cancer cell killing) between the cells due to the already investigated lytic NK-cell immunological synapse facilitating the direct secretion of lytic granules for cytotoxicity (6). The differences can be explained through the juxtacrine or paracrine signaling. Some signaling interactions are only possible by direct contact (juxtacrine signaling), while in others the signaling can also happen at higher distances between the cells (paracrine signaling).

- [4] Yang E, van Nimwegen E, Zavolan M, Rajewsky N, Schroeder M, Magnasco M, Darnell JE, Jr. 2003. Decay rates of human mRNAs: correlation with functional characteristics and sequence attributes. *Genome Res* 13: 1863-72
- [5] Blumberg A, Zhao Y, Huang YF, Dukler N, Rice EJ, Chivu AG, Krumholz K, Danko CG, Siepel A. 2021. Characterizing RNA stability genome-wide through combined analysis of PRO-seq and RNA-seq data. *BMC Biol* 19: 30
- [6] Orange JS. 2008. Formation and function of the lytic NK-cell immunological synapse. *Nat Rev Immunol* 8: 713-25

3. Cell-cell communications usually involve different numbers of cells in vivo, so is it possible to monitor the relationship of specific numbers of cells to better understand NK-Cancer cell interactions?

Response. It is indeed possible to select different numbers of each type of cell in our reported platform. However, with many different cell type and number combinations, it would be hard to predict which of the combinations best mimic the interaction biology. As such, we restricted the scope by working with cancer-NK doublets alongside the pure population of each type as control to establish a baseline of interaction and its effect on gene expression.

4. Did the authors ever compare the results of single culture and co-culture, will they exhibit a difference in kinetics and gene expression?

Response. We thank the reviewer for this insightful comment.

The Cancer-NK doublet is a type of co-culture. We compared single cancer cells (single culture) with cancer-NK doublets (co-culture) and found a small number of unconvincing differentially expressed genes (**Figure 3**). This is expected due to differences in mRNA content between NK and MDA-MB-231 cells, as explained in response to question 1 (second paragraph). It is interesting, however, to note that the MDA-MB-231 cell line, which highly expresses MHC class I molecules that inhibit NK cells, also highly expressed ICAM-1 (**differentially expressed in Figure 3**) that is involved in touching/adhesion to NK cells through LFA receptors (**Figure 6 reproduced from publication (7)**).

Figure 6. Figure reproduced from publication (7). ICAM1 is involved in NK cell adhesion.

In contrast, when we compared NK (single culture) with Cancer-NK (co-culture), we obtained reasonably strong differentially expressed genes (**Figure 7**). This can be explained due to differences in mRNA content between NK and MDA-MB-231 cells, as discussed in response to question 1 (second paragraph).

Figure 7. Gene expression comparison of doublets and single NK cells. The differential expression is clear in this analysis, highlighting the importance of the cell-cell interaction for the NK cell activation, and consequently, the gene modulation in order to tumor-killing occurrence.

[7] Garofalo C, De Marco C, Cristiani CM. 2021. NK Cells in the Tumor Microenvironment as New Potential Players Mediating Chemotherapy Effects in Metastatic Melanoma. *Front Oncol* 11: 754541

5. Did the authors ever examine tissue models? I wonder whether this approach can be used to capture inherent cell-cell interaction in tissues instead of dissociating single cells?

Response:

Did the authors ever examine tissue models: Unfortunately, our platform cannot be used to examine tissue models as we need to dissociate cells for identification and selection.

I wonder whether this approach can be used to capture inherent cell-cell interaction in tissues instead of dissociating single cells: To evaluate cell-cell interaction in tissues, methods such as 10X Visium, and Akoya multiplex DNA-tag staining can be used.

6. The results of transcriptional memory are interesting, and I wonder how can the phenomenon be reversed and whether it can be used to control the function of cells?

Response: Transcriptional memory is a potential adaptive regulatory strategy governing cellular fate decisions and allows cells to learn from previous encounters to a stimulus and respond to these cues efficiently when encountered in the future and elicit more robust and fast transcriptional responses. There are possibly different mechanisms for transcriptional memory. We speculate that transcriptional memory may be associated with longer mRNA half-lives for some genes belonging to different modules. Due to the dynamic nature of gene expression changes in the modules (**Figure 2 of our manuscript**), we hypothesize that RNA modification impacts the stability of mRNA (8) and as a result mRNA half-life. It is well known that RNA modifications such as N6-methyladenosine (m6A) and 5-methylcytosine play a significant role in RNA localization, transport, splicing, translation, and stability of mRNA. The m6A RNA modification is a dynamic and reversible post translational process (9) coordinated by multiple sets of enzymes categorized as *writers* (methyltransferases), *erasers* (demethylases) and *readers* (effector proteins). Recently, the role of m6A methylation in cell-cell communication was reported (10). The authors in this study reported m6A-mediated cell-cell communication controls planarian regeneration. In neurons, a role of m6A-RNA modification in stress response regulation is reported (11). Hence, we speculate that RNA modifications differentially enhance mRNA-stability and half-lives (12). However, analyzing these RNA modifications is beyond the scope of this manuscript. The phenomenon can potentially be reversed by interplay between writer and eraser enzymes.

We have added the above hypothesis to the discussion section of our manuscript (**line 149-154**).

- [8] Zhao BS, Roundtree IA, He C. 2017. Post-transcriptional gene regulation by mRNA modifications. *Nat Rev Mol Cell Biol* 18: 31-42
- [9] Jia G, Fu Y, He C. 2013. Reversible RNA adenosine methylation in biological regulation. *Trends Genet* 29: 108-15
- [10] Cui G, Zhou J-Y, Ge X-Y, Sun B-F, Song G-G, Wang X, Wang X-Z, Zhang R, Wang H-L, Jing Q, Zhao Y, Koziol MJ, Zeng A, Zhang W-Q, Han D-L, Yang Y, Yang Y-G. 2021. A-mediated Cell-cell Communication Controls Planarian Regeneration. *bioRxiv*: 2021.07.29.454253
- [11] Engel M, Eggert C, Kaplick PM, Eder M, Roh S, Tietze L, Namendorf C, Arloth J, Weber P, Rex-Haffner M, Geula S, Jakovcevski M, Hanna JH, Leshkowitz D, Uhr M, Wotjak CT, Schmidt MV, Deussing JM, Binder EB, Chen A. 2018. The Role of m(6)A/m-RNA Methylation in Stress Response Regulation. *Neuron* 99: 389-403 e9
- [12] Huang H, Weng H, Sun W, Qin X, Shi H, Wu H, Zhao BS, Mesquita A, Liu C, Yuan CL, Hu YC, Huttelmaier S, Skibbe JR, Su R, Deng X, Dong L, Sun M, Li C, Nachtergaele S, Wang Y, Hu C, Ferchen K, Greis KD, Jiang X, Wei M, Qu L, Guan JL, He C, Yang J, Chen J. 2018. Recognition of RNA N(6)-methyladenosine by IGF2BP proteins enhances mRNA stability and translation. *Nat Cell Biol* 20: 285-95

7. What is the throughput of this platform now? And how can it improve to achieve high-throughput analysis?

Response: The current throughput of the platform is up to 48 cell-cell interactions. Higher throughput would require a redesign of the chip with less library preparation steps on-chip.

8. In the text, there are some mistakes and missed information, such as:

(1) In page 2, line 39, “for interrogating live cell interactions at doublet resolution” should be changed to “for interrogating live-cell interactions at doublet resolution”.

Response: Thank you. We have modified the phrase in the manuscript.

(2) Please check all scale bars in the manuscript, because some of them were missed, like in Figure 1(c), Figure S1.

Response: Thank you. The figure was corrected.

RESPONSE TO COMMENTS – Reviewer 2

Reviewer #2 (Remarks to the Author):

The authors present a proof of concept for a novel approach of investigating the result of interactions between NK-cell/TNBC cell pairs. Both cell pairs as well as single cells are captured in microfluidics chambers and cultured for 13 hours. During this period the cells are continuously monitored via microscopy to determine their distance and potentially cell-contact. Cells are identified using CellTracker staining, and viability is monitored using Calcein AM. After 13 hours, microscopy is used to determine whether killing took place before cells are lysed and single-cell RNA sequencing is performed on either single cells or cell pairs as a whole. The sequencing data allowed them to differentiate the transcriptomes of killing and non-killing cell pairs and correlate these to the monitored distance. This resulted in the determination of; 1) interesting clusters (modules) of genes based on cell-cell distance, 2) transcriptional regulators of interest, 3) genes related to NK-killing events which were also found in the TCGA cohort, 4) cancer specific receptor-ligand pairs. The technique the authors propose is intriguing and very relevant to uncover single-cell heterogeneity within immune interactions. And although it has potential to amount into a useful technique, my opinion on the current state of the manuscript is that it has not yet reached the requirements needed for publication in Nature Communications Biology. Nevertheless, rewriting, which includes more elaborate explanations of the methods, addition of (already required) data, and more honest/critical discussion of limitations, might raise the manuscript to a level of quality which justifies publication. Below I will underline four major and several minor changes which in my opinion are necessary to raise the manuscripts scientific quality.

Major changes:

1. The authors indicate that the presented approach is a proof of concept. This would however require much more information and data to provide insight on how cells behave inside the microfluidic device. The first important factor is how the viability is maintained in the device, Calcein AM is incorporated so this is data that already exists. It would be very interesting to see if all cells maintain viability, including the single cells. Only then certainty can be provided if apoptosis is due to interaction instead of culture conditions, also this would underline why media change after every 5 hours is necessary. In addition to that some information should be provided on how killing events are characterized. The data processing part of the methods indicates that 10 CID`s resulted in NK cells being alone at the final time point. However also for 8 and 10 cases a single Cancer or NK cells had disappeared, can the media change have influence on this? Additionally, some data that is indispensable is the spatial information of the cells. Are the cells moving around? Can actual cell-cell attachment be identified based on microscopy images? It would greatly benefit the manuscript if a figure can be dedicated to image analysis of cell movement/distance/interaction.

Response: Please note that we have changed Tumor/Tu to Cancer in the manuscript. Citation numbers in this reply to reviewers document refer to references at the bottom of this document.

We appreciate the important observations regarding cell viability & media change, cell-killing characterization, spatial information of cells, cell-cell attachment, and image analysis of cell movement. These are important points that we will explain further.

Cell viability and media change: The viability and cell retention of different cell types in this microfluidic system has been extensively characterized in a prior publication from our group (1). As noted in this publication, the cell viability for BJ cells (human skin fibroblasts, which is an adherent cell type) after 24 hrs were around 90% (**Figure 1 in this document, which is reproduced from publication (1)**). The average cell viability for different cell types were ~90% as estimated from 20 Polaris chip runs. The media exchange frequency (every 5 hours in this study) is a feature of the system that the user can define. Since the cell culture chamber lengths are small in the microfluidic systems, we conservatively selected media exchange to be every 5 hours to replenish nutrients and clear potentially toxic metabolic byproducts that may also lower pH, similar to changing media in larger T-flasks, where the media may be changed every 2-3 days, depending on cell types. We haven't extensively characterized the frequency of medium exchange for our platform.

Figure 1. Extensive characterization of cell viability and retention. The average cell viability was ~90% as estimated from 20 Polaris chip runs

Characterization of cancer cell killing event: For the cell killing event, the cancer and NK were differentially labeled. Over time, multiple images of the chamber were captured. These time-point images were analyzed for characterization of cell killing events by obliteration (as noted in time-point 8/T8 in the image of **Figure 2**) of the blue dye signal specific for cancer cells, which is no longer detected.

Figure 2. Characterization of a killing event in all 13 time-points. Automated images were taken by the Polaris system at a one-hour interval (T1, T2, ...T13). The NK-92MI cells are shown in red color, and MDA-MB-231 cells are shown in blue color. These time-lapse images were used for cell-cell distance measurement. The culture media were replenished every five hours automatically on the Polaris system. Before single-cell RNA seq library preparation, high-resolution images were collected for each cell culture chamber using a Leica microscope system for viability tests.

Influence of media exchange: For the 8 and 10 cases of missing single cancer and NK cells, we noted instances of cell loss after media exchange. For the single cancer cells, we noted 8 cases of cell loss, out of which two disappeared during first media change and two more disappeared during second media change. For the NK cells, we noted 10 cases of cell loss, out of which seven disappeared during the first media change and two more disappeared during the second media change. For the 10 killing cases of the cancer-NK doublets, three cases showed NK loss after media change. In our prior publication, we extensively characterized cell retention during 24 hr culture using BJ cells and noted > 90% retention from eight Polaris chip runs (**Figure 1 in this document, which is reproduced from publication (1)**).

Regarding the media change, we noted distance differences on the 5th hour of incubation. Otherwise, the same media change happened again on hour 10th, but we did not observe any distance difference (**Figure 3**). So, we can see that the media change is a very still event, not causing any type of disturbance on the doublets.

Figure 3. Distance on the time-points. We can observe the distance between the cells on the 16 time points: from time zero (selection step) and one per hour. On the time points where the replacement of the media took place, two images were performed, one before and another after the replacement.

Are the cells moving around?: Yes. Some cells move around during culture, hence we measured distance over time.

Can actual cell-cell attachment be identified based on microscopy images?: Cell-cell attachments can be identified microscopically using fluorescent dyes that stain cell-cell junctions. However, we did not use those dyes in our study. In **Figure 4**, we show an NK cell (red) in direct proximity to an attached cancer cell (pink), but we did not use a dye to verify any junctional attachment between these cells.

Figure 4. Microscopic image of NK cell (red) in direct proximity to an attached cancer cell (pink).

It would greatly benefit the manuscript if a figure can be dedicated to image analysis of cell movement/distance/interaction: We have updated supplementary Figure 1 of manuscript to clarify details.

- [1] Ramalingam N, Fowler B, Szpankowski L, Leyrat AA, Hukari K, Maung MT, Yorza W, Norris M, Cesar C, Shuga J, Gonzales ML, Sanada CD, Wang X, Yeung R, Hwang W, Axsom J, Devaraju NS, Angeles ND, Greene C, Zhou MF, Ong ES, Poh CC, Lam M, Choi H, Htoo Z, Lee L, Chin CS, Shen ZW, Lu CT, Holcomb I, Ooi A, Stolarczyk C, Shuga T, Livak KJ, Larsen C, Unger M, West JA. 2016. Fluidic Logic Used in a Systems Approach to Enable Integrated Single-Cell Functional Analysis. *Front Bioeng Biotechnol* 4: 70

2. The correlation made by the authors between distance and gene expression as depicted in Figure 2B does not seem logical. Methods explain that a GxC and TxC matrix is converted into a GxT matrix. However, the gene expression is terminally profiled, and thus has no correlation with the individual distances at all 13 timepoints. The different colors as depicted in the heatmap per gene over time are depicting the correlation between the distance and gene expression. But as gene expression is the terminal value and not measured at each individual timepoint, the color might have well have depicted only the distance, because differences between timepoints are only based on distance. This also raises the question if the change at timepoint 5h (red arrow) is only due to cell movement because of media change. Additional microscopy pictures to determine or rule out that this is the case should be added. It seems much more logical to take the cell distance over time together and identify different “movement patterns” eg; “no contact” or “not contact \diamond contact \diamond release ” or “no contact \diamond contact”. Or to correlate the gene expression to the duration of cell contact. Also the authors should elaborate some more on the type of interactions they are looking for, be it juxtacrine or paracrine interactions.

Response: We thank the reviewer for the insightful comments.

Media change: Regarding the media change, we can observe that on the 5th hour of incubation, there are distance differences on some cancer-NK doublets. Also, after the same media change that occurred on hour 10, we did not observe any distance differences (**Figure 3**). The plot consists of 102 doublets of which 99 showed average distance changes $\sim 4 \mu\text{m}$ at all time points with only three chambers showing larger distance changes $> 15 \mu\text{m}$ (**Figure 5**) after the 5 hr media change.

Figure 5. Example of one out three cases where large change in distance noted after media change. Microscopic images with NK cell (red) and cancer cell (blue/pink).

Distance correlation with terminal expression: Since gene expression profiling requires cell lysis, we sacrificed the cells only at the terminal point (13th hour). It should be noted that genes that were activated at the early time points can be traced depending on their RNA half-lives even if they undergo down-regulation. This we regarded as transcriptional memory. For example, if 100 copies of a gene are present in a cell at the beginning, only 50 molecules would be left after five hours, if its half-life is five hours. Approximately 25 copies would be left after 10 hours have elapsed. We have shown that 90 genes correlate with distances throughout the time course (**Figure 2B in manuscript**). We now show that these 90 genes have relatively longer RNA half-lives (**Figure 6**).

As previously discussed in answer to Reviewer 1's question: To support a plausible hypothesis that genes belonging to different modules, shown in **Figure 2** of our manuscript, have higher overall median half-lives, we analyzed a published dataset from another group that used metabolic labeling for estimating mRNA half-lives across a number of cancer cell lines (2). In this analysis, we obtained mRNA half-life of >5000 genes from earlier published study by Blumberg, Amit, et al. in K562 cells (2) and we observed that the median half-life of genes governing cellular distances was higher than genes not belonging to the modules as depicted in the boxplot below (**Figure 6**). As a result, some of the genes that correlate with distance over the various time windows are expected to have higher mRNA half-life. In other words, mRNA half-life can be thought of as a cell's memory, which hints to its past activities.

Figure 6. Median genes half-life. Median half-life of genes governing cellular distances from the module (consist of 90 genes) was longer than genes not belonging to the modules. (Welch Two Sample t-test, p-value = 0.04867)

Addition microscopic images to characterize cell movement during media exchange:

Because the uncompiled raw images are in a very large file (approximately 2 TB), we have not posted this online. However, we show two examples of serial time point images below that cover the extremes of cell movements (minimal movement vs large movements).

Figure 2 in response to question 1 shows minimal movement of cells at T6 and T11. **Figure 5 & 7** shows large movement between cells during media change. Also note that there is a small amount of background fluorescence (faint red halo) from the bead pack.

Figure 7. Example of second case where large change in distance noted after media change. Microscopic images with NK cell (red) and cancer cell (blue/pink).

To categorize cells into the three suggested groups, we designated them as “contact” when distance between them is zero, “contact release” when they touch and move apart, and “no contact” when the two cells don’t touch each other at any time point. The boxplots below (**Figure 8**) depict the average distance across 13-time points for these three groups of cells. A heatmap of differential gene expression between these three categories is shown in **Figure 9**.

Figure 8. Cell distance categorization. The cells are categorized based on distance into contact when distance is zero micrometers and not in contact when distance is higher than zero, and the third category of cells corresponds to contact/release if cells are in contact and no contact at some point of time. Number of data points contact ($n = 20$); contact release ($n = 62$); no contact ($n = 24$)

Figure 9. Differential gene expression analysis for contact, contact release, and no contact.

Correlate the gene expression to the duration of cell contact: We correlated average physical distance across the 13-time points with gene expression profiles and obtained 27 genes based on an absolute correlation threshold value > 0.25 as shown in the bar plot (Figure 10). The 0.25 threshold lies within 1 percentile of the data (Figure 11). Notably, all these 27 genes are a subset of the 90 genes which we obtained by correlating gene expression profiles with physical distance across the 13 time points, which reinforces our data in Figure 2B of manuscript. As suggested, we hypothesized a paracrine and juxtacrine signaling among cells, correlating the observed distances with activated genes.

Figure 10. Cell distance categorization. Bar Plot from correlation average physical distance across the 13-time points with gene expression profiles and obtained 27 genes based on an absolute correlation value > 0.25.

Figure 11. Absolute correlation value distribution for threshold analysis.

Types of signaling: As previously discussed in answer to Reviewer 1, Some of the cells are in direct contact with each other at some time points while some are not (i.e. if the distance between two cells is 0, then they are touching; otherwise, they are not touching). We infer that the distance between the cells is directly related to the communication (NK activation and/or cancer cell killing) between the cells due to the already investigated lytic NK-cell immunological synapse facilitating the direct secretion of lytic granules for cytotoxicity (3). The differences can be explained through the juxtacrine or paracrine signaling. Some signaling interactions are only possible by direct contact (juxtacrine signaling), while in others the signaling can also happen at higher distances between the cells (paracrine signaling).

- [2] Blumberg A, Zhao Y, Huang YF, Dukler N, Rice EJ, Chivu AG, Krumholz K, Danko CG, Siepel A. 2021. Characterizing RNA stability genome-wide through combined analysis of PRO-seq and RNA-seq data. *BMC Biol* 19: 30
- [3] Orange JS. 2008. Formation and function of the lytic NK-cell immunological synapse. *Nat Rev Immunol* 8: 713-25

3. The authors indicate that they find hints towards the existence of transcriptional memory. This can of course be highly interesting for NK-target interactions. However, their elaboration on this is at this point too vague. Transcriptional memory as described in line 179 is “a phenomenon that allows cells to retain reversible memory to respond to similar stimuli encountered in the future” however this suggests that the NK cells have been in contact with the cancer cells before. The paper explains that only 1:1 pairing of cells is performed, and thus no serial-killing-like events can occur. In the 1st section of the methods something is mentioned about NK cell activation, which might suggest co-culture before the cell pairing. But this is not clearly described at all, and the activation data (flow cytometry data) of these NK cells is missing and should be added. The authors should elaborate much more precisely why they think transcriptional memory is occurring, and at which timepoint by which event it is initiated.

Response: We appreciate your insightful comment and apologize for not being clear about activation of NK cells.

First, the flow cytometry experiments were performed as preliminary experiments to confirm activation markers in bulk NK cells co-cultured for varying lengths of time with MDA-MB-231 cancer cells that were selected for subsequent use in the Polaris platform single cell experiments described in this manuscript. We noted the expression of NK cell activation markers as expected and this initial section titled “Standardization of NK cell activation” in the Material & Methods section should not have been included. As such, we deleted this first section from Material & Methods. We have also rewritten portions of the Materials and Methods section for further clarification, as can be seen by highlighted/tracked changes.

Transcriptional memory: As previously described, we speculated that transcriptional memory is due to longer mRNA half-lives for few genes belonging to different modules. For justification on mRNA half-life, please refer to **Figure 6** in response to question 2.

Transcriptional memory is a potential adaptive regulatory strategy governing cellular fate decisions and allows cells to learn from previous encounters to a stimulus and respond to these cues efficiently when encountered in the future and elicit more robust and fast transcriptional responses. There are possibly different mechanisms for transcriptional memory. Due to the dynamic nature of gene expression changes in the modules (**Figure 2 of our manuscript**), we

hypothesize the role of RNA modification on the stability of mRNA (4). It is well known that RNA modifications such as N6-methyladenosine (m6A) and 5-methylcytosine play a significant role in RNA localization, transport, splicing, translation, and stability of mRNA. The m6A RNA modification is a dynamic and reversible post translational process (5) coordinated by multiple sets of enzymes categorized as writers (methyltransferases), erasers (demethylases) and readers (effector proteins). Recently, the role of m6A methylation in cell-cell communication was reported (6). The authors in this study reported m6A-mediated cell-cell communication controls planarian regeneration. In neurons, a role of m6A-RNA modification in stress response regulation is reported (7). Hence, we speculate RNA modifications to differentially enhance mRNA-stability and half-lives (8) and translation of few genes that are part of our reported modules. However, analyzing those genes involved in RNA modifications is beyond the scope of this work. The phenomenon can be reversed by interplay between writers and eraser enzymes.

Why they think transcriptional memory is occurring, and at which time point by which event it is initiated: We reached this conclusion since widespread correlation was observed between cell-cell distance measurements at various time points (T0 to T13) with terminally (T13) measured gene expression levels at the doublet resolution, as illustrated in **Figure 2B in our manuscript**.

The time point at which transcriptional memory is initiated is relative to gene modules. For genes in modules M1 and M3 (**Figure 2B in the manuscript**), there is a change in correlation at T5 (time of first media change). In M1 the correlation switches from low to high correlation, while in M3 it switches from high to low correlation. For M2, the correlation is high and constant at all timepoints starting from T0. For M4, the correlation is low and constant throughout all time points.

Figure 12. Reproduced image Figure 2(B) from manuscript

We don't understand the initiation events that lead to transcriptional memory.

- [4] Zhao BS, Roundtree IA, He C. 2017. Post-transcriptional gene regulation by mRNA modifications. *Nat Rev Mol Cell Biol* 18: 31-42
- [5] Jia G, Fu Y, He C. 2013. Reversible RNA adenosine methylation in biological regulation. *Trends Genet* 29: 108-15
- [6] Cui G, Zhou J-Y, Ge X-Y, Sun B-F, Song G-G, Wang X, Wang X-Z, Zhang R, Wang H-L, Jing Q, Zhao Y, Koziol MJ, Zeng A, Zhang W-Q, Han D-L, Yang Y, Yang Y-G. 2021. A-mediated Cell-cell Communication Controls Planarian Regeneration. *bioRxiv*: 2021.07.29.454253

- [7] Engel M, Eggert C, Kaplick PM, Eder M, Roh S, Tietze L, Namendorf C, Arloth J, Weber P, Rex-Haffner M, Geula S, Jakovcevski M, Hanna JH, Leshkowitz D, Uhr M, Wotjak CT, Schmidt MV, Deussing JM, Binder EB, Chen A. 2018. The Role of m(6)A/m-RNA Methylation in Stress Response Regulation. *Neuron* 99: 389-403 e9
- [8] Huang H, Weng H, Sun W, Qin X, Shi H, Wu H, Zhao BS, Mesquita A, Liu C, Yuan CL, Hu YC, Huttelmaier S, Skibbe JR, Su R, Deng X, Dong L, Sun M, Li C, Nachtergaele S, Wang Y, Hu C, Ferchen K, Greis KD, Jiang X, Wei M, Qu L, Guan JL, He C, Yang J, Chen J. 2018. Recognition of RNA N(6)-methyladenosine by IGF2BP proteins enhances mRNA stability and translation. *Nat Cell Biol* 20: 285-95

4. The lysis and sequencing of interacting cell-pairs is incredibly interesting and holds great potential for understanding heterogeneity in immune interactions. However, it also poses enormous technical challenges, and makes it virtually impossible to determine which sequence came from which cell. In Figure S2 the authors nicely illustrate that annotations as found from microscopy data indeed correspond with gene signatures as found in the sequencing. However, it makes drawing conclusion based on discovery of genes much more difficult. In line 184-194 the authors describe three potential transcription factors as found in CID's that might be of interest. These are all factors that are commonly found to be overexpressed in breast cancer cells, making it logical that they are found in NK-breast cancer cell pairs. The reasoning why these targets are thus interesting should be much more elaborate and consider which cell pairs have had cell-cell contact and which have not.

Otherwise, these are just loose observations without conclusions. The same holds for the receptor-ligand interactions as depicted in Figure 4. If the receptor is expressed in cell 1 and the ligand is expressed in cell 2 then logically sequencing of a pair of these cells will provide a high correlation between the two. These kinds of limitation should be addressed by the author in the discussion. Additionally, the figures describing these results can be more elaborate. eg; A side by side comparison of BRCA1, THAP1 and YY1 expression per module, the results from the ShinyGO analysis next to the RCisTarget analysis, statistical analysis between "CANCER-NK/CANCER-NK" and "CANCER-NK/NK" in Figure 4 to determine if the result is actually due to a killing event, and plotting of the number of reads of both the ligand and receptors that are depicted in Figure 4.

Response:

Limitations of sequencing in pairs: We agree with the reviewer that it is not possible to determine which cell contributed which transcripts in the experiments performed in this manuscript. However, it may be possible to differentially label transcripts in future experiments (9).

Transcription factors expression per module: We obtained 90 genes across four modules by correlating terminal gene expression profiles with the physical distances among doublets across 13-time points. Based on Transcription Factors (TF) and motif enrichment analysis of the 90 genes, we predicted the regulatory role of three transcription factors, YY1, THAP1, and BRCA1. Because these genes are not among the 90 genes comprising the modules and the modules depict correlation values, it is not possible to add gene expression values of the three transcription factors as a side-by-side comparison.

Statistical analysis between “CANCER-NK/CANCER-NK” and “CANCER-NK/NK” in Figure 4 to determine if the results are due to a killing event:

We did not find a statistically significant difference for correlation between CANCER-NK/CANCER-NK and CANCER-NK/NK as shown in the table below, but we note that the small number (sample size) of NK killing events i.e. 10 events, does not provide adequate power.

	Probability
ANXA1/EGFR	0.9
HSP90AA1/EGFR	0.87
CD24/SIGLEC10	0.44

Plotting the number of reads of both the ligand and receptors that are depicted in Figure 4: We plotted transcript per million reads (TPM) values for all the three ligand and receptor pairs as shown in **Figure 13**.

Figure 13. Bar plots depicting sum of TPM values for the three ligand and receptor pairs.

- [9] Battich N, Beumer J, de Barbanson B, Krenning L, Baron CS, Tanenbaum ME, Clevers H, van Oudenaarden A. 2020. Sequencing metabolically labeled transcripts in single cells reveals mRNA turnover strategies. *Science* 367: 1151-6.

Minor changes:

5. Line 83: could the authors elaborate if heterogeneity was also observed in the single NK-cells as it was with the single Cancer cells?

Response: Unlike the single cancer cells, we did not observe cellular heterogeneity in the single NK cells, as shown from UMAP based 2D projections in **Figure 14**.

A

Figure 14. Single NK cells did not show heterogeneity. (A) refers to heterogeneity of cancer cell for comparison (supplementary Figure S3).

6. Figure 1C: By eye it seems as if cluster 1 primarily consists of NK/NK events and cluster 2 of primarily CANCER/CANCER events. Whilst the two types of CID events are distributed between. Can the authors provide per cluster the exact composition in a separate bar graph?

Response: Cluster wise exact composition of cell types in cluster 1 and cluster 2 are depicted in bar graphs in **Figure 15**. This data is now included in supplementary **Figure S4**.

Figure 15. Barplots indicating cluster wise cell-type frequencies clusters. Legend key indicates cell status at the beginning and the end of the time-course tracking (e.g., CANCER-NK/NK denotes interactions which initially consisted of both Cancer and the NK cells in a chamber, and subsequently the NK cell remained in the chamber after co-incubation due to a cell killing event).

7. Figure 1C and Figure S3A: both describe a cluster 1 and cluster 2, the article suggests that division of both clusters are due to Cancer heterogeneity, however, have both UMAP plots been made with the same data set? Or is the UMAP in Fig S3A made with only the CANCER/CANCER events?

Response: The UMAP in fig S3A was made by using CANCER/CANCER events which is a subset of Figure 1C data of manuscript.

8. Cluster 1 in Fig S3A seems to contain much more CANCER/CANCER events than cluster 1 in Fig 1C. Can the authors describe this more clearly to rule out the fact that these are just two different clustering dimensions based on different sets of events?

Response: In Fig 1C, we have used the entire dataset (n=290) to perform unsupervised clustering using the Seurat software package from R. We have 71 CANCER/CANCER events. Out of 71 CANCER/CANCER events, 18 CANCER/CANCER events are present in cluster1, and 53 are present in cluster 2. However, the FigS3A plot is based on Seurat-based unsupervised

clustering of single 71 CANCER/CANCER events alone. In this plot, there are 52 CANCER/CANCER events present in cluster1, and 19 are present in cluster 2.

9. Can the authors elaborate on how media change at t=5h causes a change in cell-cell distance and how at t=10h it does not? Perhaps by using microscopy images?

Response:

Regarding the media change, we can observe that on the 5th hour of incubation, there are distance differences on some cancer-NK doublets. Also, after the same media change that occurred on hour 10, we did not observe any distance differences (**Figure 3**). **Figure 2** in response to question 1 shows minimal movement of cells at T6 and T11. **Figure 5 & 7** shows large movement between cells during media change.

10. Can the authors elaborate on how media change can potentially affect paracrine signaling?

Response: A media change adds nutrients, removes cell metabolic by-products, and re-normalizes pH. All these factors may potentially affect paracrine signaling.

11. Can the authors add representative microscopy pictures for CIDs in different modules? To show that these modules indeed correspond to different cell distance patterns.

Response: We are unable to provide microscopy images for the modules as they represent correlations between cell-cell distances and terminal gene expression profiles of doublets ($n = 102$). There is no specific distinction of which doublets are in each module. However, below are the images for different cell interaction types (**Figure 16**).

Figure 16. Microscopic images of different cell interaction states.

12. Can the authors elaborate on the “subtle survival advantage” as observed in Figure 3? The P value is above 0.05, perhaps they can provide the actual value?

Response: We obtained a P-value of 0.0454815. This P-value is noted in legend inset of **Figure 3 in the manuscript**.

13. Line 180: authors claim that a new set of genes is expressed. This cannot be claimed if gene expression is only measured at the terminal timepoint. Can the authors elaborate on this more?

Response: We appreciate the comment. Although we measured terminal gene expression, we identified four modules that related terminal gene expression to cell-cell distances. Instead of “new set of genes”, we amended the manuscript to “different set of genes” to refer to those that contribute to modules M1 and M3 inferring that some genes may be associated with cellular interactions and eventual killing events. As this is speculative, we have not included this in the discussion section.

14. Line 181-182: Lytic synapses are hypothesized, does the obtained viability and cell-cell distance data not allow to identify these events? Or at least make them more likely?

Response: We hypothesized the existence of lytic synapses in the cases where we have NK and cancer cells touching each other and the killing event of the cancer cell occurs. Unfortunately, we did not have appropriate microscopic images to confirm lytic synapses and therefore cannot predict the likelihood of these lytic synapses. In literature, there are reports of VAV1 activation and PYK2 (protein tyrosine kinase 2) gene activity in NK cells to identify lytic synapses (doi: [10.1038/nri2381](https://doi.org/10.1038/nri2381)). However, we did not note gene expression of VAV1 and PYK2 genes in our data.

15. Line 183: could the authors elaborate on what they mean by interactions taking a drift?

Response: What we meant by interactions taking a drift is the clear changing of genes and pathways. We have updated our manuscript (lines 187-188) to clarify this confusion.

16. Figure S1: T0 is not visible at all, and it is not clear what is to be depicted there. Also, the Leica image is high-resolution, could these be cropped and zoomed to the cell pairs? And include images for several different CIDs?

Response: Thanks for your advice. We modified the supplementary figure 1.

We used the T0 image as a selection step. We could quickly review whether the cells were loaded on the microfluidic chip or not.

In supplementary figure 1, we adjusted the dimension of the Leica image to the cell pairs. However, we only collected the Leica image at T13 to maintain and assess cell viability before single-cell RNA sequencing, so as not to perturb the cells during culture conditions.

We thank the reviewer for his/her meticulous reading of our manuscript and providing such pertinent comments for its improvements.

Reviewers' comments:

Reviewer #1 (Remarks to the Author):

This manuscript reported a cell-cell interaction study that combines live capture of different cell types, co-incubation, time-lapse imaging, and gene expression profiling of doublets using a microfluidic integrated fluidic circuit (IFC). The physical distances between cells can be directly linked to downstream transcriptomics changes. Highly coordinated gene expression profiles were identified as a result of dynamic changes in the physical distance of interacting NK and cancer cells. Upon further investigation, these transcripts were confirmed to have overall higher RNA half-lives reinforcing transcriptional memory as a key regulatory strategy of cells and increased coordination among some specific ligand-protein pairs were traced. This microfluidic system may serve as an attractive approach to study the immune-oncology cellular interactions at the single-cell level. The following are comments on this manuscript.

1. The biological systems in vivo usually involve multiple cells and rely on the interactions between cells to coordinate cell signaling and regulate cell functions, so what's the meaning of studying cell-cell interactions at the single-cell level? It is better to explain that in the introduction of manuscript.
2. In Figure 1C, the NK/NK group mainly exhibits in cluster1 and the Cancer/Cancer group exhibits in cluster1 and cluster2. However, the NK/NK and Cancer/Cancer groups are very different cell types. Why did Cancer/Cancer group and NK/NK group exhibit in the same cluster1? The UMAP in Fig S3A was made by using Cancer/Cancer events which is a subset of Figure 1C data of manuscript. It is recommended to made UMAP using only Cancer/Cancer cells to confirm that the separation of Cancer/Cancer group into cluster1 and cluster2 is due to the heterogeneity of cancer cells.
3. The average cell viability for different cell types was ~90% after 24-hour incubation as shown in Figure1 in response to comments of reviewer 2. The single cell viability raises the question: the cancer cell apoptosis in NK killing events is due to cell-cell interaction or culture conditions? As shown in Figure 1D, differentially expressed genes distinguished killing and non-killing events. However, considering the cell viability, the NK killing events may be less than 10. It is difficult to perform statistical analysis of the transcriptomic data obtained from less than 10 killing events.
4. As shown in Figure 2B, the correlation between distance and gene expression changed after 5-hour incubation and culture medium change. Since authors profiled the gene expression of cells only at the terminal point (13th hour), the genes activated at the early time points (such as 5th hour) were traced based on transcriptional memory. To confirm the correlation change is due to cell-cell interaction or medium change, it is recommended to directly compare the distance and gene expression after 5-hour incubation with and without cell culture medium change.
5. What's the pairing efficiency and throughput of cell doublets in microfluidic integrated fluidic circuit (IFC)? It is recommended to briefly introduce the workflow of generating cell doublets in IFC with a schematic diagram in Supporting Information.

Reviewer #2 (Remarks to the Author):

The authors have adequately addressed all of my concerns and as such I endorse publication of this work.

RESPONSE TO COMMENTS – Reviewer 1

Reviewer #1 (Remarks to the Author):

This manuscript reported a cell-cell interaction study that combines live capture of different cell types, co-incubation, time-lapse imaging, and gene expression profiling of doublets using a microfluidic integrated fluidic circuit (IFC). The physical distances between cells can be directly linked to downstream transcriptomics changes. Highly coordinated gene expression profiles were identified as a result of dynamic changes in the physical distance of interacting NK and cancer cells. Upon further investigation, these transcripts were confirmed to have overall higher RNA half-lives reinforcing transcriptional memory as a key regulatory strategy of cells and increased coordination among some specific ligand-protein pairs were traced. This microfluidic system may serve as an attractive approach to study the immune-oncology cellular interactions at the single-cell level. The following are comments on this manuscript.

1. The biological systems in vivo usually involve multiple cells and rely on the interactions between cells to coordinate cell signaling and regulate cell functions, so what's the meaning of studying cell-cell interactions at the single-cell level? It is better to explain that in the introduction of manuscript.

We appreciate the comment of Reviewer 1.

Traditional sequencing methods can only identify cell populations, analyzing the average of the signals within each group of cells, but the heterogeneity in tumor cells is lost. Single-cell sequencing technologies can perfectly detect the heterogeneity of the cells. Further, in doublets of cells, it is possible to find specific markers, as well as cell ligands and correlate them with the anti-tumor response drawn in each doublet [1].

Using this workflow, it is possible to add more cells and study their interactions. This workflow reports interaction at doublet resolution to resolve heterogeneity with possibilities to study more cells

As suggested by the reviewer, this further justification was added to the introduction.

[1] Tang, X., Huang, Y., Lei, J., Luo, H., & Zhu, X. (2019). The single-cell sequencing: new developments and medical applications. *Cell & bioscience*, 9, 53.

2. In Figure 1C, the NK/NK group mainly exhibits in cluster1 and the Cancer/Cancer group exhibits in cluster1 and cluster2. However, the NK/NK and Cancer/Cancer groups are very different cell types. Why did Cancer/Cancer group and NK/NK group exhibit in the same cluster1? The UMAP in Fig S3A was made by using Cancer/Cancer events which is a subset of Figure 1C data of manuscript. It is recommended to made UMAP using only Cancer/Cancer cells to confirm that the separation of Cancer/Cancer group into cluster1 and cluster2 is due to the heterogeneity of cancer cells.

We appreciate the observation of Reviewer 1.

Though Cancer/Cancer event is a subset of data in Fig1C, FigS3A was made using only the Cancer/Cancer cells by subjecting only cancer/cancer cells to unsupervised clustering using the Seurat based pipeline, revealing heterogeneity among cancer cells. We have now moved Fig S3A to the main manuscript as panel D in Fig 1, so it is more accessible to the readers. We thank the reviewer for that suggestion.

Figure S3A (now Fig 1 panel D). UMAP-based visualization of single cancer cells showing heterogeneity in the cancer cell line.

To confirm that the MDA-MB-231 cells and the NK cells are indeed different cell types, even though they clustered within the same group 1, we performed additional analyses. We used the top 100 differential genes from cluster 1, Figure 1c and subjected them to Enrichr to perform gene list enrichment analyses.

Figure 1c. UMAP-based visualization of cells.

Indeed, the MDA-MB-231 cell line is the top term in the Diseases/Drugs category of Enrichr and Natural killer and basal cells are the top terms in the Cell Type category of Enrichr, as shown in the figure below.

Diseases/Drugs category of Enrichr

Cell Type category of Enrichr

Further, cluster 1 is dominated by NK/NK cells in comparison to Cancer/Cancer cell as shown in the barplot below.

3. The average cell viability for different cell types was ~90% after 24-hour incubation as shown in Figure1 in response to comments of reviewer 2. The single cell viability raises the question: the cancer cell apoptosis in NK killing events is due to cell-cell interaction or culture conditions? As shown in Figure 1D, differentially expressed genes distinguished killing and non-killing events. However, considering the cell viability, the NK killing events may be less than 10. It is difficult to perform statistical analysis of the transcriptomic data obtained from less than 10 killing events.

Thank you for this observation. Due to small sample size, we purposely implemented the limma-voom method [1] to determine differential expression analyses, as this method can handle small sample sizes.

Below are some quotes from Law *et al.* Limma-voom manuscript [1]

“Borrowing information between genes is a crucial feature of the genome-wide statistical methods, as it allows for gene-specific variation while still providing reliable inference with small sample sizes. The normal-based empirical Bayes statistical procedures can adapt to different types of datasets and can provide exact type I error rate control even for experiments with a small number of replicate samples”

“A key advantage of the limma pipelines is that they provide accurate type I error rate control even when the number of RNA-seq samples is small.”

[1] Law, Charity W., et al. "voom: Precision weights unlock linear model analysis tools for RNA-seq read counts." *Genome biology* 15.2 (2014): 1-17.

4. As shown in Figure 2B, the correlation between distance and gene expression changed after 5-hour incubation and culture medium change. Since authors profiled the gene expression of cells only at the terminal point (13th hour), the genes activated at the early time points (such as 5th hour) were traced based on transcriptional memory. To confirm the correlation change is due to cell-cell interaction or medium change, it is recommended to directly compare the distance and gene expression after 5-hour incubation with and without cell culture medium change.

We appreciate the suggestion of the reviewer. As suggested, we correlated gene expression profiles with the physical distance before 5 hours (without culture medium change) and after 5 hours (with culture medium change), as shown in the heatmaps below. We therefore obtained 55 genes in the case without medium change and 32 genes when culture medium was changed. We found 7 genes (PHF10, FABP5, ASB7, MORN2, EIF2D, TNFRSF21 and GALNT11) that were common in these groups. To select genes, we used a threshold of 0.24 which lies in the 1 percentile of the data. Therefore, we conclude that the gene expression programs are due to cell-cell interactions rather than the media change.

5. What's the pairing efficiency and throughput of cell doublets in microfluidic integrated fluidic circuit (IFC)? It is recommended to briefly introduce the workflow of generating cell doublets in IFC with a schematic diagram in Supporting Information.

The pairing efficiency is 85%. It is calculated by number of total doublets/ (24 expected doublets per IFC * number of IFCs).

The throughput of cell doublets per IFC is up to 48 doublets. However, we added 12 single cells for each of the two cell types to every IFC plate. Therefore, each IFC would have a maximum of 24 doublets.

We have prepared a new schematic diagram added to Figure S1 to better illustrate the cell doublets generation for our Polaris experiments.

The cells were cultured in bulk, stained, and pipetted on to the Polaris IFC in bulk. The selection of the candidates for single cell started inside the IFC. Cells with higher viability (by Calcein AM stain) were selected and incubated either as single NK or cancer cells or as NK-cancer cell doublets.

We thank the reviewer for this very careful and detailed review. We hope we have answered all questions to your satisfaction.

We have been very appreciative of the inputs from both reviewers and feel that this has significantly improved our manuscript.

REVIEWERS' COMMENTS:

Reviewer #1 (Remarks to the Author):

The revised manuscript has addressed the reviewer's concerns. The manuscript is recommended for publication in the journal.